# *NCAPG* Regulates Myogenesis in Sheep, and SNPs Located in Its Putative Promoter Region Are Associated with Growth and Development Traits

**DOI:** 10.3390/ani13203173

**Published:** 2023-10-11

**Authors:** Zehu Yuan, Ling Ge, Pengwei Su, Yifei Gu, Weihao Chen, Xiukai Cao, Shanhe Wang, Xiaoyang Lv, Tesfaye Getachew, Joram M. Mwacharo, Aynalem Haile, Wei Sun

**Affiliations:** 1Joint International Research Laboratory of Agriculture and Agri-Product Safety of Ministry of Education of China, Yangzhou University, Yangzhou 225009, China; yuanzehu@yzu.edu.cn (Z.Y.); gl1024winnie@163.com (L.G.); supengwei4901@163.com (P.S.); feilin133082@163.com (Y.G.); 18552133709@163.com (W.C.); cxkai0909@163.com (X.C.); 007121@yzu.edu.cn (S.W.); dx120170085@yzu.edu.cn (X.L.); 2International Joint Research Laboratory in Universities of Jiangsu Province of China for Domestic Animal Germplasm Resources and Genetic Improvement, Yangzhou University, Yangzhou 225009, China; 3College of Animal Science and Technology, Yangzhou University, Yangzhou 225009, China; 4International Centre for Agricultural Research in the Dry Areas, Addis Ababa 999047, Ethiopia; t.getachew@cgiar.org (T.G.); j.mwacharo@cgiar.org (J.M.M.); a.haile@cgiar.org (A.H.); 5“Innovative China” “Belt and Road” International Agricultural Technology Innovation Institute for Evaluation, Protection, and Improvement on Sheep Genetic Resource, Yangzhou 225009, China

**Keywords:** sheep growth and development, myoblasts, NCAPG, SNP

## Abstract

**Simple Summary:**

Screening for polymorphisms in the promoter region of a functional gene is an effective way to identify useful markers for improving sheep growth and development. Non-SMC condensin I complex subunit G (*NCAPG*) is a candidate gene linked with sheep growth and development. Its explicit role in muscle development is still unclear, and markers in *NCAPG*’s promoter region have not been explored yet. The goal of this study was to investigate the direct role of *NCAPG* in regulating myogenic development and the differentiation of myoblasts and explore potential markers in its promoter region in relation to sheep growth and development traits. To achieve this goal, cell proliferation and differentiation after RNA interference with *NCAPG* were investigated in embryonic myoblasts. In addition, the genetic markers in the promoter region of *NCAPG* were scanned, and association analysis between the markers and sheep growth and development traits was carried out. The results suggest that interfering with *NCAPG* inhibits the proliferation and differentiation of myoblasts. Five variants detected in the promoter region of *NCAPG* were significantly (*p* < 0.05) associated with sheep growth and development traits. These results provide direct evidence of *NCAPG* regulating myogenesis and provide useful genetic markers to increase the efficacy of the selection of sheep growth and development traits.

**Abstract:**

Previously, *NCAPG* was identified as a candidate gene associated with sheep growth traits. This study aimed to investigate the direct role of *NCAPG* in regulating myogenesis in embryonic myoblast cells and to investigate the association between single-nucleotide polymorphisms (SNPs) in its promoter region and sheep growth traits. The function of *NCAPG* in myoblast proliferation and differentiation was detected after small interfering RNAs (siRNAs) knocked down the expression of *NCAPG*. Cell proliferation was detected using CCK-8 assay, EdU proliferation assay, and flow cytometry cell cycle analysis. Cell differentiation was detected via cell immunofluorescence and the quantification of myogenic regulatory factors (MRFs). SNPs in the promoter region were detected using Sanger sequencing and genotyped using the improved multiplex ligation detection reaction (iMLDR^®^) technique. As a result, a notable decrease (*p* < 0.01) in the percentage of EdU-positive cells in the siRNA-694-treated group was observed. A significant decrease (*p* < 0.01) in cell viability after treatment with siRNA-694 for 48 h and 72 h was detected using the CCK-8 method. The quantity of S-phase cells in the siRNA-694 treatment group was significantly decreased (*p* < 0.01). After interfering with *NCAPG* in myoblasts during induced differentiation, the relative expression levels of MRFs were markedly (*p* < 0.05 or *p* < 0.01) reduced compared with the control group on days 5–7. The myoblast differentiation in the siRNA-694 treatment group was obviously suppressed compared with the control group. SNP1, SNP2, SNP3, and SNP4 were significantly (*p* < 0.05) associated with all traits except body weight measured at birth and one month of age. SNP5 was significantly (*p* < 0.05) associated with body weight, body height, and body length in six-month-old sheep. In conclusion, interfering with *NCAPG* can inhibit the proliferation and differentiation of ovine embryonic myoblasts. SNPs in its promoter region can serve as potential useful markers for selecting sheep growth traits.

## 1. Introduction

Sheep are well adapted to diverse environments and are raised principally for their meat, milk, and fiber. These products could meet global needs for meat and essential nutrients for an increasing population around the world, especially in developing countries or regions. Growth and development traits, including body weight, body length, and body height, have economic importance in sheep production because these traits reflect production efficiency. Therefore, improving sheep growth and development traits is important for increasing the economic returns for farmers and the meat supply for customers. Genomic selection is one of the most effective ways to improve economic traits [1]. However, the economic returns of sheep genomic selection are still limited. The costs of genotyping and computation are still unaffordable for large-scale sheep populations in developing countries [2]. Marker-assisted selection is an alternative way to increase the rate of genetic improvement, which could reduce the time required to improve sheep growth at an acceptable cost. Thus, finding key candidate genes and then designing genetic markers for the genetic improvement of growth and development traits is essential for the sheep industry.

By utilizing the advancements in high-throughput sequencing and genotyping, researchers have effectively employed genome-wide association analysis (GWAS) using single-nucleotide polymorphisms (SNPs) across the whole genome to discover SNPs and important genes associated with growth and development traits in sheep populations from various backgrounds [3,4,5,6,7,8,9,10,11,12,13,14,15,16,17,18,19,20,21,22,23,24]. These studies provide an important theoretical basis for further investigating functional genes and designing genetic markers for the genetic improvement of growth and development traits. Among these GWAS studies, several studies with relatively large population sizes have identified a striking candidate gene called non-SMC condensin I complex subunit G (*NCAPG*) on ovine chromosome 6 associated with sheep growth and development traits [4,7,8]. Beef cattle GWAS results suggest that *NCAPG* is linked with an average daily gain [25]. A meta-analysis of GWAS for cattle stature identified that *NCAPG* may regulate body size in mammals [26]. In beef research, the expression of *NCAPG* in the fetal longissimus muscle was significantly greater than that in adults [27], and the abundance of *NCAPG* was significantly correlated with the average daily gain trait [25]. Recently, a study in cattle revealed that *NCAPG* could regulate myogenesis in the fetal stage [28]. Nevertheless, explicit evidence of *NCAPG* regulating sheep growth and development is still scarce, especially in muscle development.

The development of skeletal muscles is a crucial indicator of a sheep’s growth and development characteristics since it has been estimated using computerized tomography (CT) scans that muscles make up more than 60% of a sheep’s body weight [19]. Muscle development can usually be classified into two stages, namely the prenatal and postnatal stages. Muscle development during the prenatal period is crucial as it determines the number of muscle fibers, which scarcely increases after birth [29,30,31]. Myoblast proliferation, differentiation, and fusion determine prenatal myogenesis [32,33]. To date, the role of *NCAPG* in the myogenesis of sheep myoblasts is still largely unknown.

When a gene’s function is determined, the genetic markers linked with economic traits in the gene’s regulatory regions, e.g., its promoter region, can be explored. For example, in Romney sheep, SNPs in the promoter region of the myostatin (*MSTN*) were significantly related to growth and carcass traits [34]. Similarly, in Hu sheep, SNPs in the promoter region of the melanocortin-4 receptor (*MC4R*) were linked with body conformation traits [35]. Despite the importance of the SNPs in the promoter region of a functional gene, no studies investigating the association between SNPs in *NCAPG*’s promoter region and growth and development traits in sheep have been implemented.

The goal of this study was to investigate the direct role of *NCAPG* in regulating myogenic development and differentiation of myoblasts and explore potential SNPs in its promoter region in relation to sheep growth and development traits. To achieve this goal, cell proliferation and differentiation after RNA interference with *NCAPG* were investigated in myoblasts derived from fetal ovine muscle tissue. In addition, SNPs in the promoter region of *NCAPG* (from transcription start site to upstream 2000 bp of transcription start site) were detected by Sanger sequencing and genotyped by the improved multiplex ligation detection reaction (iMLDR^®^) method. The association analysis between SNPs and sheep growth and development traits was implemented. This study could provide direct evidence on *NCAPG* in regulating myogenesis and provide useful genetic markers in *NCAPG*’s promoter region to increase the selection efficacy for sheep growth and development traits.

## 2. Materials and Methods

### 2.1. Ethical Statement

Animal experiments in the current study underwent censorship and received approval from Yangzhou University’s Experimental Animal Ethical Committee (approval number 202103279, approval date 8 March 2021).

### 2.2. Sample and Phenotypic Data Collection

One experimental pregnant ewe of Hu sheep used for the primary embryonic myoblasts’ isolation was provided by Suzhou Taihu Dongshan Sheep Industry Development Co., Ltd. (Suzhou 215000, Jiangsu Province, China). The pregnant ewe was slaughtered, and the fetal *longissimus dorsi* muscle was obtained from its descendants. Then, the fetal *longissimus dorsi* muscle was conserved in an insulated bucket and brought back to the laboratory for the subsequent isolation of cells.

Suhu meat sheep (Figure 1) is a breeding population that was constructed by our group. It derives from the selective breeding of white Dorper and Hu sheep. All Suhu meat sheep were raised under the same management conditions in Xuzhou Suyang Sheep Industry Co., Ltd. (Xuzhou 215000, Jiangsu Province, China). Six types of tissue samples, namely *longissimus dorsi* muscle, heart, liver, spleen, lung, and kidney, were collected from Suhu sheep. In total, three fetuses (around day 85) from two pregnant ewes, three five-day-old male newborn lambs, and three one-year-old male sheep were used for tissue sample collection. All tissue samples were collected within 30 min with three triplicates after the sheep were slaughtered. Samples were placed in liquid nitrogen and stored in a refrigerator set at −80 °C for long-term preservation.

In the current study, the 2nd generation of Suhu meat sheep (Figure 1) was used for phenotypic data collection. Four traits, namely body weight, body height, body length, and shin circumference, representing sheep growth and development were measured at born, one month, two months, three months, and six months following a published protocol [36]. A total of 539 blood samples with phenotypic data were collected, including 190 male and 349 female lambs. Genomic DNA was extracted from blood using a conventional phenol-chloroform method. Double-distilled water was used to dissolve the DNA samples, which were then stored in a refrigerator at a temperature of −20 °C.

### 2.3. Isolation of Sheep Embryonic Myoblasts

Primary embryonic myoblasts were successfully isolated in our previous study [37]. Briefly, embryonic myoblasts were obtained following three steps: (1) primary embryonic myoblast isolation, (2) primary embryonic myoblast purification, and (3) embryonic myoblast identification. Firstly, primary embryonic myoblasts were isolated from the fetal *longissimus dorsi* tissue by using the collagenase and trypsin combined digestion differential adhesion methods [38]. Secondly, embryonic myoblasts were purified by using the cell suspension method [37]. Thirdly, primary embryonic myoblasts were identified by using quantitative analysis of myogenic regulatory factors (MRFs) after inducing differentiation. Finally, 0.25% trypsin was used to digest myoblasts, and then they were frozen in liquid nitrogen for further experiments. The cells were cultured in a growth medium (GM) with a temperature of 37 °C and 5% CO_2_.

### 2.4. Plasmid Construction and Small Interfering RNA Synthesis

The coding sequence (CDS) of *NCAPG* was amplified from embryonic myoblasts’ complementary DNA (cDNA). Primers of CDS used for plasmid construction designed using the CE Design online tools (http://www.vazyme.com, accessed on 2 October 2023) and Primer Premier 5 [39] are documented in Table 1. The pcDNA3.1(+) plasmid was digested using *Hind* III and *Bam*H I restriction enzymes, resulting in linearization. Next, the PCR product was inserted into the linearized vector according to the guidelines provided by the ClonExpress^®^ II One Step Cloning Kit (Vazyme Biotech Co., Ltd., Nanjing 210000, Jiangsu Province, China). Finally, the constructed recombinant plasmid was named pcDNA3.1(+)-*NCAPG*. To confirm pcDNA3.1(+)-NCAPG, Sanger sequencing was performed by a commercial sequencing provider (Tsingke Biotechnology Co., Ltd., Nanjing 210000, Jiangsu Province, China).

Small interfering RNAs (siRNAs) specifically targeting ovine *NCAPG* were designed and synthesized by a commercial service provider (GenePharma Pharmaceutical Technology Co., Ltd., Suzhou 215000, Jiangsu Province, China). Meanwhile, a negative control (NC) was also synthesized. All the synthesized siRNA sequences were documented in Table 2.

### 2.5. Cell Transfection and Induction of Differentiation

The procedures of cell transfection and induction of differentiation followed our previous work [37]. In short, once the cells reached a confluence level of 50–60%, the transfection was carried out by using the jetPRIME transfection reagent (Polyplus transfection, Strasbourg, Illkirch, France). Following transfection for 24–48 h, embryonic myoblasts were induced to differentiate in vitro when GM was changed into a differential medium (DM, consisting of 98% high-glucose Dulbecco’s Modified Eagle Medium (DMEM) and 2% horse serum). Afterward, the cells were gathered for additional RNA extraction.

### 2.6. Total RNA Extraction, Reverse Transcription, and RT-PCR

TRIzol reagent (TIANGEN, Beijing, China) was used to extract total RNA from tissues and cells. A NanoReady spectrophotometer (Life Real, Hangzhou, China) was used to measure RNA concentration. In addition, 1% agarose gels were used to detect RNA integrity and contamination. All total RNA samples were stored in a −80 °C refrigerator until use.

FastKing gDNA Dispelling RT Super Mix (TIANGEN, Beijing, China) was utilized for reverse transcription according to the guidelines provided by the manufacturer. RT-PCR was conducted on the CFX96 Connect™ Real-Time System (BIO-RAD, Hercules, CA, USA) using 2 × TSINGKE Master qPCR Mix (SYBR Green I) (Tsingke, Beijing, China) following the manufacturer’s guidelines. To normalize the gene expression, the abundance of the housekeeping gene *GAPDH* was utilized. The commonly used 2^−∆∆Ct^ method was used to calculate gene abundance [40]. The primers were designed by Primer Premier 5 [39] and are displayed in Table 3.

### 2.7. Cell Proliferation Detection

In the current study, cell proliferation after interference with *NCAPG* was detected by Cell Counting Kit-8 (CCK-8) assay, 5-Ethynyl-20 -deoxyuridine (EdU) proliferation assay, and flow cytometry for cell cycle analysis.

#### 2.7.1. CCK-8 Assay

The CCK-8 assay was performed using a CCK-8 kit (Vazyme, Nanjing, China) according to the instructions provided by the manufacturer. In short, embryonic myoblasts were seeded into the 96-well plates and were cultured in GM. Subsequently, cells were transfected using the methods documented in Section 2.5 for a duration of 24 h. Afterward, 10 µL of cck-8 solution was gently added into each well. Cells were incubated at 37 °C in darkness for 2–3 h. Finally, the optical density (OD) value at 450 nm representing the relative cellular activity at 0 h, 24 h, 48 h, and 72 h was detected by using a microplate reader (Tecan, Männedorf, Switzerland).

#### 2.7.2. EdU Proliferation Assay

The EdU cell proliferation kit (RIBOBIO, Guangzhou, China) was utilized to carry out the EdU proliferation assay. Initially, embryonic myoblasts were transfected for 24–48 h. Then, the EdU staining assay was carried out following the manufacturer’s guidelines, and an inverted fluorescence microscope (Nikon, Minato, Tokyo, Japan) was used to take fluorescent photos. Finally, cell numbers in each image were counted using Image J (National Institutes of Health, Bethesda, MD, USA). The calculation involved using a formula to determine the percentage of EdU-positive cells, which was obtained by dividing the number of EdU-stained cells by the number of Hoechst-stained cells and multiplying the result by 100%.

#### 2.7.3. Flow Cytometry for Cell Cycle Analysis

Flow cytometry was performed using a Cell Cycle kit (Beyotime, Shanghai, China). In short, trypsin was employed for cell digestion. After embryonic myoblasts were transfected for 24–48 h, the cells were gathered in a centrifuge tube with a capacity of 1.5 mL. Subsequently, 70% ethanol, which had been pre-cooled at 4 °C, was added into the centrifuge tube to preserve the cells for a duration of 12 h. Afterward, propidium (PI) staining was employed to color the cell samples in a dark environment at a temperature of 37 °C for 30 min. In the end, the DNA content of the cells was detected using Modfit (Version 3.1) (Topsham, ME, USA) on a BD LSRFortessa flow cytometer (BD, Franklin Lakes, NJ, USA).

### 2.8. Cell Differentiation Detection

Cell differentiation was detected in the present study using cell immunofluorescence and quantification of myogenic regulatory factors (MRFs).

#### 2.8.1. Cell Immunofluorescence Staining

The detailed immunofluorescence staining procedure can be found in a previous study [37]. The main experimental steps were as follows:(1)Discard the cell culture medium; rinse 1–2 times with 1 × PBS on a destaining shaker; add 500 μL cell fixative to each well and incubate at room temperature for 15–30 min;(2)Discard the fixative and rinse with 1 × PBS destaining shaker 3–4 times, 5 min/time; add 500 μL of 0.5% Triton X-100 to each well, and incubate with destaining shaker for 20–30 min to enhance cell membrane permeability;(3)Discard the permeabilization solution and rinse with 1 × PBS decolorizing shaker 3–4 times, 5 min/time; add 500 μL of freshly prepared 10% goat serum working solution to each well, and incubate at 37 °C in the dark for 60 min;(4)Discard the blocking solution, add 300 μL of freshly prepared primary antibody working solution (MyHC, 1:200) to each well, and incubate overnight at 4 °C in the dark;(5)Discard the primary antibody working solution, and rinse with 1 × PBS decolorizing shaker 3–4 times, 5 min/time; add 300 μL of freshly prepared secondary antibody working solution (Goat Anti-Rabbit IgG (H + L)) to each well, 1:1000), incubate at 37 °C in the dark for 60 min;(6)Discard the secondary antibody working solution and rinse with 1 × PBS destaining shaker 3–4 times, 5 min/time;(7)Add 300 μL of DAPI working solution to each well, and incubate at room temperature in the dark for 3–5 min; rinse with 1 × PBS decolorizing shaker 3–4 times, 5 min each time, and immediately place it under an inverted fluorescence microscope to take pictures and observe.

#### 2.8.2. MRF Quantification

Given that myogenesis is primarily controlled by MRFs, which include Myogenic Differentiation 1 (*MYOD*), Myogenic Factor 5 (*MYF5*), Myogenin (*MYOG)*, and Myogenic factor 6 (*MYF6*, also referred to as *MRF4*), in the present investigation, all of these MRFs were measured using RT-qPCR according to the procedure outlined in Section 2.6. The primers used to quantify MRFs were adopted from a previous study [37].

### 2.9. Sequencing and Genotyping of Putative NCAPG Promoter Region

Five pairs of primers used to amplify putative *NCAPG* promoter region were designed using Primer Premier 5 [39] (Table 4). This region spans from the transcription start site to 2000 bp upstream of the transcription start site. Fifty individuals were mixed as a DNA pool with each genomic DNA 50 ng/μL. A total of 11 DNA pools were constructed. Then, PCR was conducted using DNA pools as templates. The detailed PCR protocol can be found in a previous study [37]. The PCR products were sequenced by Sanger sequencing on an ABI3730 (Tsingke Biotechnology Co., Ltd., Beijing, China). Sanger sequencing results were analyzed to find putative SNPs using Chromas software (Technelysium Pty Ltd., Helensville, Australia). Potential SNPs of each individual were genotyped using iMLDR (Genesky Biotechnologies Inc., Shanghai, China), which is an improved ligase-based multiplex SNP genotyping system. The detailed procedures can be found in a previous study [41]. The ability of SNPs to change the binding affinity of transcription factors to their binding sites was predicted using a web interface (https://azifi.tz.agrar.uni-goettingen.de/agreg-snpdb, accessed on 2 October 2023).

### 2.10. Statistical Analysis

The significance of RT-qPCR results was tested using a *t*-test (between two groups) or one-way analysis of variance (ANOVA, more than two groups) in SPSS 25.0 (SPSS, Inc., Chicago, IL, USA).

Population genetic parameters, including allele frequencies, minor allele frequency (MAF), expected heterozygosity (He), observed heterozygosity (Ho), polymorphic information content (PIC), and Hardy–Weinberg equilibrium statistic of all SNPs, were calculated using the snpReady R package [42]. The formulas to calculate these parameters are documented in the manual of snpReady [42].

Association analysis between SNPs and phenotypes was implemented by using SPSS 25.0 (SPSS, Inc., Chicago, IL, USA) to fit a linear model. The model was as follows:Phe = µ + Gnt + Gnd + err
where Phe is the individual phenotype value, µ is the population mean value, Gnt is the genotype effect, Gnd is the gender effect, and err is the random error. Differences in mean phenotypes among the three genotypes were present in the least square mean value and were tested by using the LSD test. *p* < 0.05 denotes significance. *p* < 0.01 denotes high significance.

## 3. Results

### 3.1. NCAPG Was Highly Expressed in Embryonic Sheep Muscle

To investigate the temporal and cellular expression of *NCAPG* at different growth and development stages, the RT-qPCR method was used to detect its relative expression levels in the heart, liver, spleen, lungs, kidney, and *longissimus dorsi* tissues of 120-day-old fetal sheep, 5-day-old newborn lambs, and 6-month-old Suhu sheep. The results indicated that the abundance of *NCAPG* in the *longissimus dorsi* muscle, heart, lung, and kidney tissues of embryotic sheep was significantly higher than that in 5-day-old sheep and 6-month-old sheep (*p* < 0.05) (Figure 2A). In the embryotic sheep *longissimus dorsi* muscle, the average abundance of *NCAPG* was nearly 1300 times higher than that in 5-day-old sheep and 6-month-old sheep. Moreover, the extensive analysis of time-series detection at the cellular level indicated that *NCAPG* exhibited a progressively decreased expression pattern at various time points (*p* < 0.05) following the initiation of myoblast differentiation (Figure 2B). The findings reveal that the abundance of *NCAPG* could have a crucial impact on the growth and development of embryonic sheep muscle.

### 3.2. The Expression of NCAPG Was Reduced by siRNA Knockdown

To elucidate the regulating role of *NCAPG* in proliferation and differentiation in sheep embryonic myoblasts, small interfering RNA (siRNA) was synthesized and overexpression recombinant plasmids were constructed to change the abundance of *NCAPG* at the cellular level. Firstly, siRNA targeting *NCAPG* was transfected into fetal sheep myoblasts to detect the relative expression of *NCAPG*. The results showed that siRNA-694, siRNA729, and siRNA-2259 significantly reduced the relative expression level of *NCAPG* (*p* < 0.01) (Figure 3A), and the interference efficiency exceeded 70%, so siRNA-694, siRNA729, and siRNA-2259 could be used in subsequent experiments. Considering that the interference efficiency of siRNA-694 was higher than that of the other two siRNAs, siRNA-694 was used for further experiments. Then, the recombinant plasmid targeting *NCAPG* was transfected into fetal sheep myoblasts, and the results showed that the relative expression levels of *NCAPG* in the pcDNA3.1(+) and pcDNA3.1(+)-NCAPG groups were similar (Figure 3B). Thus, the subsequent functional verification experiment of *NCAPG* was implemented using RNA interference.

### 3.3. Interfering with NCAPG Inhibits the Proliferation of Sheep Fetal Myoblasts

To further investigate whether the abundance of *NCAPG* affects the proliferation of fetal sheep myoblasts, siRNA-694 was transfected into cells, followed by conducting EdU staining, the CCK-8 assay, and flow cytometry analysis to examine cell cycle distribution. The EdU staining findings indicated a notable decrease in the percentage of EdU-positive cells in the siRNA-694 treated group compared to the control group (*p* < 0.01) (Figure 4A,B). The CCK-8 method revealed a significant decrease (*p* < 0.01) in cell viability, as indicated by the reduced OD values (Figure 4C), after treatment with siRNA-694 for 48 h and 72 h, suggesting that interference with *NCAPG* weakened cell viability. The findings from the detection of the cell cycle indicated a significant decrease (*p* < 0.01) in the quantity of S-phase cells in the siRNA-694 treatment group compared to the control group (Figure 4D,F). This suggests that interference with *NCAPG* greatly impeded the cellular proliferation process.

### 3.4. Interfering with NCAPG Hinders the Differentiation of Sheep Fetal Myoblasts

To investigate whether interference with *NCAPG* also impacts the differentiation of fetal sheep myoblasts, siRNA-694 was transferred into myoblast cells. Interfering with *NCAPG* in induced-differentiation myoblasts on days 5–7 was found to markedly reduce the relative expression levels of *MYOD1*, *MYOG*, *MYF5*, and *MRF4* (*p* < 0.05 or *p* < 0.01) (Figure 5A–D). The findings from MyHC indirect immunofluorescence analysis on the 7th day of differentiation induction (Figure 5E) indicate that the myoblast differentiation of the siRNA-694 treatment group was considerably suppressed compared to that of the control group. Moreover, there was a decrease in the number of fused myotubes, and a lower level of differentiation was observed in the siRNA-694 treatment group, suggesting that interference with *NCAPG* inhibits myoblast differentiation. Based on the above results, the expression changes of *NCAPG* can significantly affect the proliferation and differentiation of fetal sheep myoblasts.

### 3.5. Five SNPs in the Putative Promoter Region of NCAPG

Pooled Sanger sequencing was used to search for the potential SNPs in the NCAPG promoter region. The sequencing peak map showed that a total of five SNPs were detected in the 2000 bp of the *NCAPG* promoter region (Figure 6 and Appendix A). After the quality assessment of the five SNPs by Genesky Biotechnologies Inc. (Shanghai, China), the genotyping results of the five SNPs were subjected to negative control quality control and repeated quality control. The results showed that the negative control had no signal. The results of repeated samples were consistent, indicating that the genotyping results were reliable. Finally, five SNPs were successfully genotyped (Table 5). The results suggest that three genotypes were detected at each SNP. All five SNPs deviate from Hardy–Weinberg equilibrium (*p* < 0.05). The polymorphic information content values of these five SNPs were 0.36, 0.36, 0.36, 0.36, and 0.33, respectively. All five SNPs have moderate polymorphism (Table 5). Further, the functional prediction of five SNPs suggests that SNP3, SNP4, and SNP5 may change the binding affinity of transcription factors to their binding sites (Appendix A).

### 3.6. SNPs in NCAPG Promoter Region Associated with Sheep Growth and Developmental Traits

The associations between the five SNPs and four traits at four development stages were detected. SNP1, SNP2, SNP3, and SNP4 were significantly associated with all traits except body weight measured at birth and one month of age (Table 6 and Appendix A). SNP5 was significantly (*p* < 0.05) associated with body weight, body height, and body length in six-month-old sheep. For SNP1, individuals carrying the *AA* genotype were characterized by the highest values of investigated traits. For SNP2, individuals carrying the *TT* genotype were characterized by the highest values of investigated traits. For SNP3, individuals carrying the *CC* genotype were characterized by the highest values of investigated traits. For SNP4, individuals carrying the *TT* genotype were characterized by the highest values of investigated traits. For SNP5, individuals carrying the *AA* genotype were characterized by the highest values of investigated traits.

## 4. Discussion

Several GWAS studies of sheep with a relatively large population size have found that *NCAPG* can be used as a candidate gene for sheep growth and development traits such as body weight and body size [4,7,8]. An expression quantitative trait loci (eQTL) study in sheep detected a very low expression level of *NCAPG* in the liver and muscle in a 7- to 8-month-old sheep population [43]. This suggests the expression level of *NCAPG* was tissue- or (and) time-specific. Thus, the *NCAPT* expression levels in six tissues and three development stages were analyzed, and it was found that *NCAPG* is highly expressed in embryotic sheep muscle (Figure 2A). This is consistent with the results in beef research where the expression level of *NCAPG* in the *longissimus dorsi* muscle tissue of 135-day-old fetal muscle was significantly higher than the relative expression level of *NCAPG* in adult muscle [25]. Further, the time-series detection analysis of abundance at the cellular level revealed that *NCAPG* showed a gradually downregulated expression pattern at different time points after the induction of myoblast differentiation (Figure 2B). The protein level of *NCAPG* found in cattle research [28] showed a similar expression pattern to our result. All these pieces of evidence suggest that *NCAPG* plays different roles in the different stages of the muscle cell development process.

Cell proliferation occurs as a result of the combination of cell growth with regular “G1-S-M-G2” cell cycles to produce many diploid cell progeny. RNA interference at the cellular level was implemented to investigate the role of *NCAPG* in myoblast proliferation. In the current study, the results suggest that knocking down *NCAPG* expression could inhibit cell proliferation (Figure 4). This is consistent with the result in cattle research where *NCAPG* knockdown prolonged the procession of myoblast mitosis [28]. In human research, *NCAPG* could promote the proliferation of many types of cells, such as hepatocellular carcinoma cells [44,45], colorectal cancer cells [46], and pulmonary artery smooth muscle cells [47]. All these pieces of evidence suggest that *NCAPG* could increase proliferation in various types of cells. Previous research suggests that NCAPG as a mitosis-associated chromosomal condensation protein plays an important role in the appropriate separation of sister chromatids [48]. In the current study, the number of cells in the S phase was significantly increased and the number of cells in the M phase was increased (not significantly) when *NCAPG* was inhibited (Figure 4). In cattle, *NCAPG* inhibition prolonged the prometaphase and metaphase of proliferating myoblasts [28]. Considering the function conservation of *NCAPG* in growth and development traits across mammals [26], we confidently speculate that *NCAPG* regulates proliferation through prometaphase and metaphase. All these pieces of evidence suggest that *NCAPG* could increase differentiation in various types of cells.

Differentiation is different from proliferation [37,38]. In the current study, two pieces of evidence support the inhibition of *NCAPG* inhibiting myogenic differentiation in myoblasts (Figure 5). A human cancer meta-analysis suggests that *NCAPG* has a relationship with cell differentiation [49]. In cattle, *NCAPG* inhibited myogenic differentiation in myoblasts [28]. All these pieces of evidence suggest that *NCAPG* could increase proliferation in various types of cells. Previous studies suggest that chromatin accessibility could regulate embryonic muscle development in pigs [50] and cattle [51]. In cattle research, it was found that *NCAPG* may adjust chromatin accessibility [28]. Thus, we speculate that *NCAPG* regulates myogenic differentiation by regulating chromatin accessibility.

Screening for SNPs in the putative promoter region of functional genes is an effective way to identify useful genetic markers for animal breeding. In this study, all identified SNPs in the promoter region of *NCAPG* were associated with at least one growth and development trait (Table 6 and Appendix A). Consistent with the results in sheep research, a number of effects of SNPs located in the promoter regions on growth [35,52], carcass [34,53], meat quality [34,53], and reproductive traits [54,55] were reported. Our finding is also consistent with the results from other livestock species showing that *SNPs* found in the promoter region are significantly associated with economically important traits [56,57,58,59]. SNPs in the promoter region may influence promoter activity, transcription factor binding, DNA methylation, and histone modifications [60]. For example, an SNP in the *NR5A2* promoter region could regulate litter size in Hu sheep by increasing the promoter activity [55]. Similar research has reported that a novel SNP in the *IGF1* promoter region could regulate the litter size of Yunshang black goats by increasing transcription-promoting activity [59]. Thus, these five SNPs may regulate the expression of *NCAPG* and influence sheep growth and development by adjusting the activity of the promoter.

## 5. Conclusions

In summary, the results of the current study have revealed that interfering with *NCAPG* inhibits the proliferation of sheep fetal myoblasts and differentiation. Five SNPs in the promoter region of *NCAPG* were significantly associated with sheep growth and development traits. Considering the functional prediction of SNPs and *the p-value* of association analyses, SNP3 might be the most important mutation for selecting sheep growth and development traits. These results could provide direct evidence of *NCAPG* regulating the proliferation and differentiation of embryonic myoblasts and provide useful genetic markers to increase the efficacy of the selection of sheep growth and development traits.

## Figures and Tables

**Figure 1 animals-13-03173-f001:**
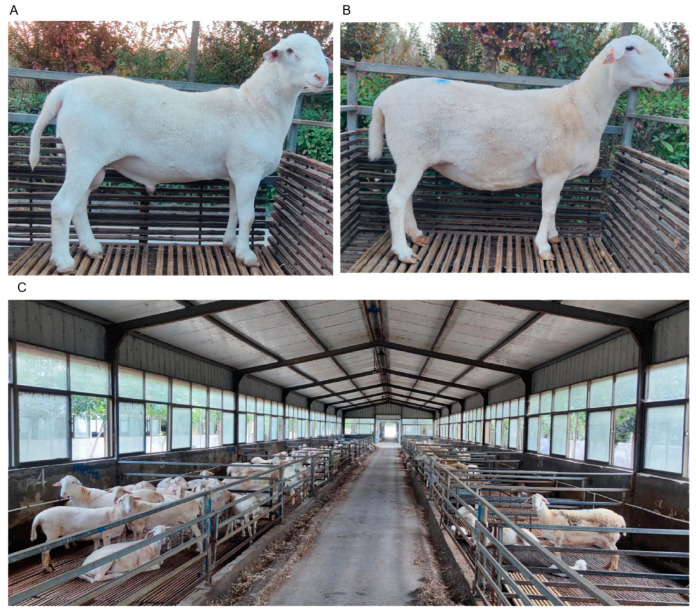
Suhu meat sheep. (**A**) Male; (**B**) female; (**C**) population.

**Figure 2 animals-13-03173-f002:**
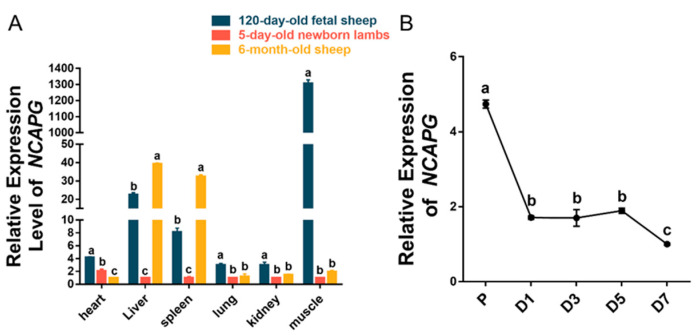
The expression of *NCAPG* was quantified in six different tissues during three growth stages, as well as in fetal sheep myoblasts across four stages of differentiation. (**A**) The relative expression levels of *NCAPG* were detected in six tissues of 120-day-old fetal sheep, 5-day-old newborn lambs, and 6-month-old sheep using RT-qPCR. (**B**) The relative expression levels of NCAPG were detected in fetal sheep myoblasts at proliferation (P) and on the 1st, 3rd, 5th, and 7th days (D1, D3, D5, and D7) of differentiation. Mean values with different letters denote significance (*p* < 0.05).

**Figure 3 animals-13-03173-f003:**
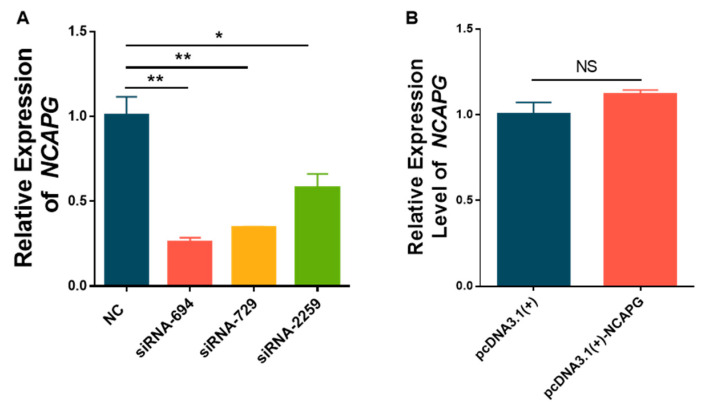
Validation of the efficiency of siRNA and recombinant plasmid targeting *NCAPG*. (**A**) The efficiency of siRNA targeting *NCAPG* was detected in fetal sheep myoblasts by RT-qPCR. (**B**) The efficiency of pcDNA3.1(+)-NCAPG targeting *NCAPG* was detected in fetal sheep myoblasts by RT-qPCR. * *p* < 0.05, ** *p* < 0.01, NS *p* > 0.05.

**Figure 4 animals-13-03173-f004:**
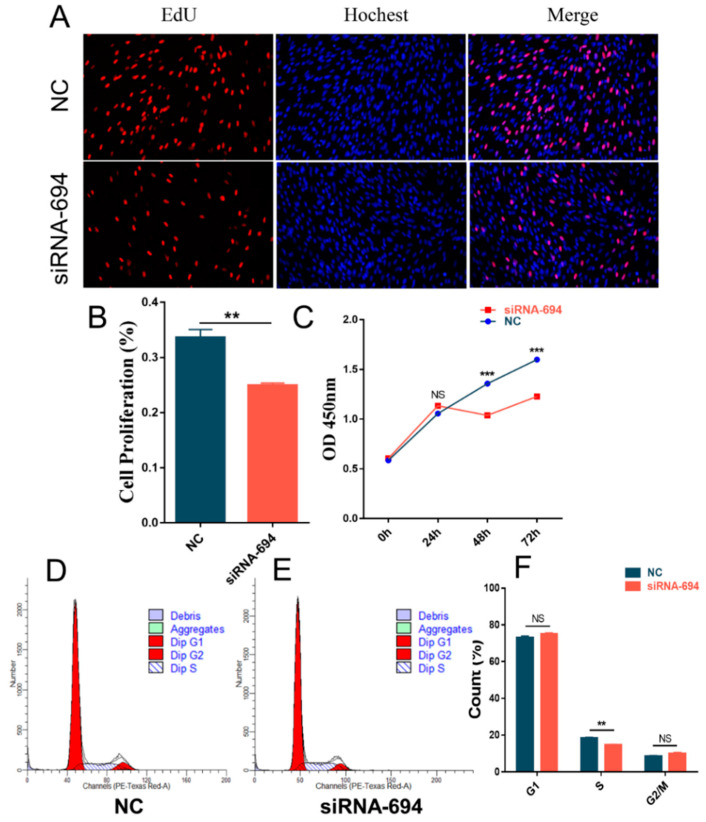
Functional validation of the effect of *NCAPG* interference on the proliferation of fetal sheep myoblasts. (**A**) Results of EdU staining of fetal sheep myoblasts transfected with siRNA-694 (100×). (**B**) Proliferation rate of fetal sheep myoblasts transfected with siRNA-694. (**C**) The OD values of fetal sheep myoblasts transfected with siRNA-694 at 0 h, 24 h, 48 h, and 72 h were detected by CCK-8. (**D**) The distribution of fetal sheep myoblasts transfected with a negative control at different stages was detected by flow cytometry. (**E**) The distribution of fetal sheep myoblasts transfected with siRNA-694 at different stages was detected by flow cytometry. (**F**) Proportion of the number of fetal sheep myoblasts transfected with siRNA-694 at different stages of the cell cycle. ** *p* < 0.01, *** *p* < 0.001, NS *p* > 0.05, *n* = 6 biological replicates.

**Figure 5 animals-13-03173-f005:**
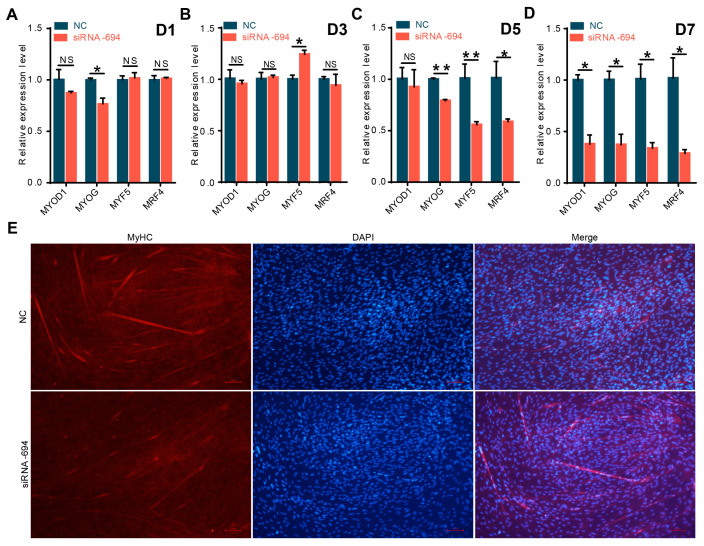
Functional analysis of the effect of *NCAPG* interference on differentiation of fetal sheep myoblasts. A-D: The relative expression levels of four MRFs detected in fetal sheep myoblasts after siRNA-694 treatment on the 1st (**A**), 3rd (**B**), 5th (**C**), and 7th (**D**) days (D1, D3, D5, and D7) of differentiation. (**E**) MyHC immunofluorescence staining was observed in fetal sheep myoblasts transfected with siRNA-694 using a fluorescence inverted microscope (100×). * *p* < 0.05, ** *p* < 0.01, NS *p* > 0.05, *n* = 3 biological replicates.

**Figure 6 animals-13-03173-f006:**
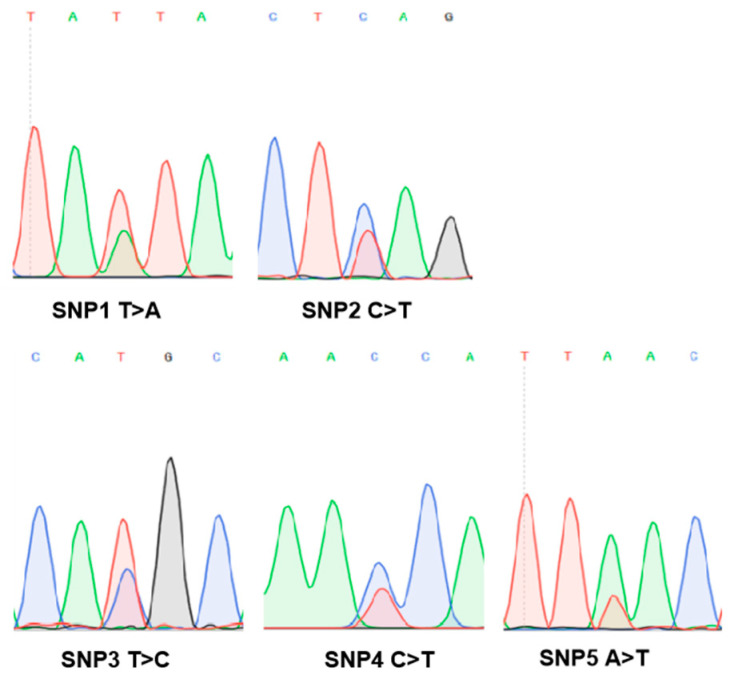
Potential SNPs in the promoter region of *NCAPG*.

**Table 1 animals-13-03173-t001:** Primers used in plasmid construction.

Primer Name	Sequence (5′-3′)	Product Length
NCAPG-F	Gggagaccc^1^aagct^2^ggctagcATGGGGAAGGAGAAGAGACTGC	3054 bp
NCAPG-R	agtggatccgagctcggtaccCTAACTTGTATCTTCATTGAGAAATTGTG	

^1^ The lowercase letters at the 5′-end of the primer sequence indicate the 15 bp sequence at the end of the linearized pcDNA3.1(+) vector. ^2^ The lowercase underlined letters indicate the restriction sites, and the uppercase letters indicate the insertion fragment sequences.

**Table 2 animals-13-03173-t002:** Sequence information for small interfering RNAs (siRNAs) targeting ovine NCAPG and negative control (NC).

Name	Primer Name	Sequence (5′-3′)
NCAPG	siRNA-694	F: GCGCCAUCAGCAAAGACUUTT
		R: AAGUCUUUGCUGAUGGCGCTT
	siRNA-729	F:GCGCACCAAGGAUGUGAAATT
		R: UUUCACAUCCUUGGUGCGCTT
	siRNA-2259	F:GGUCAGCAGCAGGAUUCUUTT
		R:AAGAAUCCUGCUGCUGACCTT
	NC	F:UUCUCCGAACGUGUCACGUTT
		R:ACGUGACACGUUCGGAGAATT

**Table 3 animals-13-03173-t003:** Specific primers used for RT-qPCR.

Name	Primer Name	Sequence (5′-3′)	GenBank Accession
*NCAPG*	*NCAPG*-F	AGAGACTGCTGCCGATTAAGG	XM_027970895.2
	*NCAPG*-R	ACCTGTTTTGTCATCCACCGAG	
*GAPDH*	*GAPDH*-F	TCTCAAGGGCATTCTAGGCTAC	NM_001190390.1
	*GAPDH*-R	GCCGAATTCATTGTCGTACCAG	

**Table 4 animals-13-03173-t004:** Specific primers used to amplify the promoter regions.

Primer Name	Sequence (5′-3′)	Amplified Region	Product Length
NCAPG-F	CAAGCATATTTCATGTACTCTTAA	R1	880 bp
NCAPG-R	TATTGTTTTCATGTTCTAAGGTTAT		
NCAPG-F	AGTTATGGGAAAAAGCAGCTTCC	R2	360 bp
NCAPG-R	TGTTTTCAACAAAGGAGAACACAGA		
NCAPG-F	GGCTGTAATTATGGAGATGTTGATT	R3	480 bp
NCAPG-R	TCTAATTACTTAAACCTCTCCCCAC		
NCAPG-F	CAGACTCTTCACCAGCAATTTCACT	R4	370 bp
NCAPG-R	AGTGAGTACCCTGTGGAGATTCG		
NCAPG-F	AGTCCCTGTTCTCACCCATGG	R5	435 bp
NCAPG-R	ACGAAGCCTGTTCACGTTCG		

**Table 5 animals-13-03173-t005:** Population genetic parameters and Hardy–Weinberg equilibrium test of SNPs.

SNP	MAF ^1^	He ^2^	Ho ^3^	PIC ^4^	Chisq Value	*p* ^5^
SNP1	0.39	0.48	0.59	0.36	29.357	6.02 × 10^8^
SNP2	0.38	0.47	0.57	0.36	23.915	1.01 × 10^6^
SNP3	0.39	0.47	0.58	0.36	24.562	7.20 × 10^7^
SNP4	0.38	0.47	0.57	0.36	23.915	1.01 × 10^6^
SNP5	0.29	0.42	0.48	0.33	12.256	4.64 × 10^4^

^1^ MAF: minor allele frequency; ^2^ He: expected heterozygosity; ^3^ Ho: observed heterozygosity; ^4^ PIC: polymorphic information content; ^5^
*p* value of Hardy–Weinberg equilibrium test.

**Table 6 animals-13-03173-t006:** Overall *p* values of association analysis between SNPs and phenotypes.

Age	Traits	SNP1	SNP2	SNP3	SNP4	SNP5
Birth	Body weight	0.329	0.350	0.371	0.350	0.365
	Body height	0.005	0.005	0.005	0.005	0.646
	Body length	0.038	0.038	0.039	0.038	0.498
	Shin circumference	0.018	0.019	0.020	0.019	0.540
One month	Body weight	0.306	0.310	0.308	0.310	0.296
	Body height	0.004	0.004	0.005	0.004	0.598
	Body length	0.011	0.011	0.012	0.011	0.547
	Shin circumference	0.008	0.009	0.010	0.009	0.533
Two months	Body weight	0.020	0.025	0.025	0.025	0.947
	Body height	0.004	0.004	0.004	0.004	0.985
	Body length	0.004	0.005	0.005	0.005	0.984
	Shin circumference	0.006	0.006	0.007	0.006	0.898
Three months	Body weight	0.010	0.013	0.014	0.013	0.853
	Body height	0.002	0.002	0.002	0.002	0.802
	Body length	0.001	0.002	0.002	0.002	0.895
	Shin circumference	0.007	0.009	0.009	0.009	0.637
Six months	Body weight	0.007	0.015	0.016	0.015	0.041
	Body height	0.001	0.002	0.002	0.002	0.014
	Body length	0.001	0.003	0.003	0.003	0.032
	Shin circumference	0.001	0.003	0.003	0.003	0.056

## Data Availability

No new data were created or analyzed in this study.

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
