# Peer review of "NCAPG Regulates Myogenesis in Sheep, and SNPs Located in Its Putative Promoter Region Are Associated with Growth and Development Traits"

_animals, 2023, doi:10.3390/ani13203173_

Round 1
Reviewer 1 Report
NCAPG is a candidate gene linked with sheep grow and development from previous GWAS. This manuscript investigated the direct role of NCAPG in regulating myogenesis in embryonic myoblast cells by siRNAs knocking down the expression of NCAPG and the association between SNPs in its promoter region and sheep growth traits. Introduction is well explained why this study was implemented. Materials and Methods were easy to follow. Conclusion was supported by results. These results could provide direct evidence on NCAPG to regulate myogenesis and provide useful genetic markers to increase the selection efficacy of sheep growth and development traits. I have several minor comments to improve this manuscript.
In title. The the promoter region of NCAPG is a potential promoter region. So, please rephrase this title accordingly.
In line136.-137 Please explicit indicate how many generations were selected in “Suhu” meat sheep breeding population.
In Figure 1. Please provide a population photo of “Suhu” meat sheep.
In Figure 5. Please revise the gene name in correct format. Such as ,MyoD1.
In conclusion. Please explicit which SNP was the most import one for selecting sheep growth and development traits.
Author Response
Reviewer1
Q1: NCAPG is a candidate gene linked with sheep grow and development from previous GWAS. This manuscript investigated the direct role of NCAPG in regulating myogenesis in embryonic myoblast cells by siRNAs knocking down the expression of NCAPG and the association between SNPs in its promoter region and sheep growth traits. Introduction is well explained why this study was implemented. Materials and Methods were easy to follow. Conclusion was supported by results. These results could provide direct evidence on NCAPG to regulate myogenesis and provide useful genetic markers to increase the selection efficacy of sheep growth and development traits. I have several minor comments to improve this manuscript.
A1: Thank you very much for your valuable time and constructive comments for improving our manuscript. We have revised our manuscript according to your valuable suggestions.
Q2: In title. The promoter region of NCAPG is a potential promoter region. So, please rephrase this title accordingly.
A2: Thank you very much for your valuable suggestion. We have revised the title as “NCAPG Regulates the Myogenesis in Sheep and SNPs Located in Its Putative Promoter Region are Associate with Growth and Development Trait” according to your and another reviewer’s suggestion. Please see title.
Q3: In line136.-137 Please explicit indicate how many generations were selected in “Suhu” meat sheep breeding population.
A3: Thank you very much for your comment. All these animals selected for implemented association analysis were selected from 2nd generation of “Suhu” meat sheep breeding population. We have added this information in text: “In the current study, 2nd generation of Suhu meat sheepwere used for phenotypic data collection.” Please see line151.
Q4: In Figure 1. Please provide a population photo of “Suhu” meat sheep.
A4: Thank you very much for your comment. A population photo of “Suhu” meat sheep has been added in Figure 1. Please see Figure 1.
Q5: In Figure 5. Please revise the gene name in correct format. Such as, MyoD1.
A5: Thank you very much for your careful work. We have revised the gene name in correct format in Figure 5. Please see Figure 5.
Q6: In conclusion. Please explicit which SNP was the most import one for selecting sheep growth and development traits.
A6: Thank you very much for your suggestion. Combing the functional prediction of SNP and P value of association analyses, SNP3 might be the most important mutation for selecting sheep growth and development traits. Please see lines 489-491
Author Response
Reviewer2
Q1: Submitted manuscript is very interesting, performed analysis are complexed and methods are well chosen. Obtained results could be applied in practice. I recommend it for publish in Animals, but the text needs few corrections.
A1: Thank you very much for your valuable time and constructive comments for improving our manuscript, especially for improving our English phrasing. We have incorporate your valuable comments in our manuscript.
Q2: Authors should give more data about detected polymorphism e.g position, type of substitution, if it is possible also rs. See Ensembl to check if SNPs were earlier detected or are new.
A2: Thank you very much for your constructive comments. The information SNP were listed as below. The information of all these five SNPs were also documented in Table S1.
SNP |
Position |
Ref > Alt |
Mutation type (Novel or Existed) |
Distance to TSS (bp) |
SNP1 |
Chr6: 42090126 |
T>A |
rs424493003 |
-1726 |
SNP2 |
Chr6: 42090844 |
C>T |
rs159958117 |
-1008 |
SNP3 |
Chr6:42091061 |
T>C |
rs423376306 |
-791 |
SNP4 |
Chr6:42091075 |
C>T |
rs417096593 |
-777 |
SNP5 |
Chr6:42091109 |
T > A |
rs430255987 |
-743 |
Q3: I think that it will be valuable for the text to predict function of detected SNP e.g. analysis of TFBS.
A3: Thank you very much for your constructive comments. The ability of SNPs to change the binding affinity of transcription factors to their binding sites was predicted by a web interface (https://azifi.tz.agrar.uni-goettingen.de/agreg-snpdb, accessed on October 2nd, 2023). The results of functional prediction were listed as below and were also documented in Table S1.
SNP |
Core similarity score |
Matrix similarity score |
Sequence |
Consequence |
SNP1 |
NA |
NA |
NA |
NA |
SNP2 |
NA |
NA |
NA |
NA |
SNP3 |
1 |
0.867 |
aacatgCACGTttgtaa |
Score-Change |
1 |
0.866 |
aacacgCACGTttgtaa |
Score-Change |
|
1 |
0.945 |
ctaaACAACacgca |
Gain of TFBS |
|
1 |
1 |
CACGCa |
Gain of TFBS |
|
SNP4 |
1 |
0.898 |
gtttgtaAATCAggcct |
Gain of TFBS |
SNP5 |
1 |
1 |
aATTAA |
Gain of TFBS |
See below for minor hints, please:
Q4: I think the title "NCAPG Regulates the Myogenesis in Sheep and SNPs Located in Its Promoter Region Are Associated with Growth and Development Traits" sounds better
A4: Thank you very much for your constructive suggestion. We have revised the title as “NCAPG Regulates the Myogenesis in Sheep and SNPs Located in Its Putative Promoter Region Are Associated with Growth and Development Traits”. Please see title.
Q5: Line 26 " Screen genetic markers in promoter..." → "Screen for polymorphism in promoter..."
A5: Thank you very much for your careful work. We have revised “Screen genetic markers” as “Screen for polymorphism”. Please see line 26.
Q6: Line 27 "for improving the sheep growth and development"
A6: Thank you very much for your careful work. We have deleted “the” in the manuscript. Please see line 27.
Q7: Line 28 - "grow" → "growth"
A7: Thank you very much for your careful work. We have revised “grow” as “growth”. Please see line 28.
Q8: Line 29 - "little markers in NCAPG’s promoter region were explored" → "markers in NCAPG’s promoter region were not explored yet"
A8: Thank you very much for your careful work. We have revised "little markers in NCAPG’s promoter region were explored" as “markers in NCAPG’s promoter region were not explored yet”. Please see line 30.
Q9: Line 34 - " region linked with sheep growth and development traits" → "region in relation to sheep growth and development traits"
Q9: Thank you very much for your careful work. We have revised " region linked with sheep growth and development traits" as “region in relation to sheep growth and development traits”. Please see line 32.
Q10: Line 37 - "Five markers in the promoter" → "Five variants/SNPs detected in the promoter"
A10: Thank you very much for your careful work. We have revised " Five markers in the promoter” as "Five variants/SNPs detected in the promoter". Please see line 37.
Q11: Line 42 - "embryotic" → "embryonic"
A11: Thank you very much for your careful work. We have revised “embryotic” as “embryonic”. Please see lines 42-43.
Q12: Line 46 - "NCAPG"→ "NCAPG"
A12: Thank you very much for your careful work. We have italicized NCAPG in main text. Please see line 46.
Q13: Line 46 - "Flow Cytometry" → "flow cytometry"
A13: Thank you very much for your careful work. We have revised "Flow Cytometry" as "flow cytometry". Please see line 46-47.
Q14: Line 48 - "was scanned" → "was detected"
A14: Thank you very much for your careful work. We have revised "scanned" as "detected".
Q15.Line 49, 119, 178, 391 - "sanger" → "Sanger"
A15: Thank you very much for your careful work. We have revised "sanger" as "Sanger". Please see line 48.
Q16: Line 49 - add full name of method in bracket "improved multiplex ligation detection reaction"
A16: Thank you for your careful work. Full name of iMLDR have been added in text. Please see line 49.
Q17: Line 52 - "decrease" → "decreased"
A17: Thank you for your careful work. We have revised “decrease” as “decreased”. Please see line 53.
Q18: Line 57 - "were" → "was"
A18: Thank you for your careful work. We have revised “were” as “was”. Please see line 58.
Q19: Line 59, 88 - "Ovine" → "ovine"
A19: Thank you for your careful work. We have revised “Ovine” as “ovine”. Please see line 60, 88.
Q20: Line 59, 478 - i think word "hamper" should be replaced by "inhibit" in the whole text
A20: Thank you very much for your valuable suggestion. “hamper" has been replaced by "inhibit" in the whole text.
Q21: Line 60 - "promoter region can be serve"
A21: Thank you very much for your careful work. ‘be’ has been deleted in the text. Please see line 59, 355, 386, 462, 464 and 487.
Q22: Line 65 - "sheep are well"
A22: Thank you very much for your careful work. ‘are’ was added between “sheep” and “well”.
Q23: Line 65 - rewrite this part, please " convert short herbage mainly to meat"
A23. Thank you for your suggestion. We have rephrased this sentence as “Sheep are well adapted to diverse environment and are raised principally for their meat, milk and fiber”. Please see line 65.
Q24: Line 68 - delete "mainly"
A24. Thank you for your suggestion. We have deleted “mainly” in the text. Please see line 68.
Q25: Line 87 - "gene, called/named"
A25: Thank you for your suggestion. “called” has been added after “gene”. Please see line 87.
Q26: Line 98 - explain "CT" shortcut
A26: Thank you for your careful work. Full name of CT (computerized tomography) has been add in the text. Please see line 98.
Q27: Line 101 - "during the prenatal period"
A27: Thank you for your careful work. “the” has been deleted. Please see line 101.
Q28: Line 109 - "MC4R"
A28: Thank you for your careful work. “MC4R” has been italic. Please see line 109.
Q29: Line 111 - "little" → "no"
A29: Thank you for your suggestion. “little” has been changed into “no”. Please see line 111
Q30: Line 113 - " The goal/s of this study was/were to investigate the direct role of NCAPG in regulating myogenic development and differentiation of myoblasts and explore potential SNPs markers in its promoter region linked with in relation to sheep growth and development traits."
A30: Thank you for your valuable suggestion. “markers” has been deleted and “linked with” has been replaced by “in relation to”. Please see line 115.
Q31: Line 115-119 - in my opinion these sentences could be omitted, however if authors will decide to leave them, correct this one "In addition, the SNPs in promoter region of NCAPG (from transcription start site to upstream 2000 bp of transcription start site) was scanned were searched/detected by Ssanger sequencing and genotyped by iMLDR® method".
A31: Thank you very much for your suggestion. Yes, this sentence can be omitted. Considering this sentence can provide detailed information for potential reader, we decided to keep this sentence and revised this sentence according to your valuable suggestion. Please see line 119-120.
Q32: Line 148 - "Figure 1. Suhu sheep of six-month-old. A: male Suhu sheep; B: female Suhu sheep" →"Figure 1. Six-month-old Suhu sheep A: male; B: female"
A32: Thank you very much for your valuable suggestion. According to your and reviewer1’s suggestion, we added a population photo of Suhu Meat sheep in Figure 1. Thus, the legend of Figure 1 was revised as “Figure 1. Suhu meat sheep. A: male; B: female; C: population”. Please see Figure 1.
Q33: Line 150 - "represent" → "representing"
A33: Thank you for your valuable suggestion. We have revised “represent” as “representing”. Please see line 153.
Q34: Line 178 - "Sanger sequencing was performed by a commercial sequencing provider" - is not easier to run electrophoresis and check construct length by DNA ladder? Was the whole insert sequenced or fragment? If whole - more than one sequencing reaction was run, because maximum read is 800-1000bp, however insert length was 3054bp.
A34: Thank you very much for your professional comments. In our study, whole insert sequence was sequenced by three Sanger sequencing reaction.
Q35: Line 199 - explain shortcut "DMEM"
A35: Thank you very much for your careful work. The full name of DMEM is “Dulbecco's Modified Eagle Medium”. We have added it in the main text. Please see line 201.
Q36: Line 212 - which tool was used to primers design? Add info, please.
A36: Thank you very much for your useful suggestion. The primers were designed by Primer Premier 5. We have added this info in main text. Please see line 215.
Q37: Italicize gene names in Tables!!!
A37: Thank you very much for your careful work. We have italicized gene names in Table 1.
Q38: Line 237 - " Flow Cytometry was performed using a cell cycle kit (Beyotime, Shanghai, China)" → "Flow cytometry was performed using a Cell Cycle Kit (Beyotime, Shanghai, China)."
A38: Thank you very much for your careful work. We have revised “Flow Cytometry was performed using a cell cycle kit (Beyotime, Shanghai, China)" as "Flow cytometry was performed using a Cell Cycle Kit (Beyotime, Shanghai, China)." Please see line 241.
Q39: Line 250 - "The detailed Immunofluorescence" → "The detailed immunofluorescence"
A39: Thank you very much for your careful work. We have revised “Immunofluorescence” as “immunofluorescence”. Please see line 254.
Q40: Line 274 - give full name of genes too
A40: Thank you very much for your valuable suggestion We have added full names for Myogenic Differentiation 1 (MYOD), Myogenic Factor 5 (MYF5), Myogenin (MYOG), and Myogenic factor 6 (MYF6). Please see line 277-279.
Q41: Line 277 - "premiers" → "primers"
A41: Thank you very much for your careful work. We have revised "premiers" as "primers". Please see line 281.
Q42: Line 279 - "A total of five pairs of primers used to amplify putative NCAPG promoter region were designed by Primer Premier 5 [39] (Table 4)" sounds better
A42: Thank you very much for your valuable comment. We have rephrased this sentence as "A total of five pairs of primers used to amplify putative NCAPG promoter region were designed by Primer Premier 5 [39] (Table 4)" according to your valuable suggestion. Please see line 283-284.
Q43: A total of five pairs of primers vs 2000bp. I wonder why authors applied 5 primer pairs. In my opinion 3 pairs (3x800=2400, even 880 as visible in Table 4. 3x880=2640bp) are enough to determine 2000bp sequence.
A43: Thank you very much for your professional comments. Yes, theoretically, 3 pairs primers are enough to determine 2000bp sequence. In the currently, in order to easily design the primers, we used five pairs primers.
Q44: Line 295 vs 307 - repeated data
A44: Thank you very much for your careful work. We have deleted “P < 0.05 denotes significant. P < 0.01 denotes highly significant” In line 295. Please see line 301.
Q45: Line 335 - "siRNA-134, siRNA-406, and siRNA-694 significantly reduced the relative expression" then other siRNAs (siRNA-694, siRNA729 and siNRA-2259) were used in subsequent experiments?
A45: Thank you very much for your careful work and sorry for our carefulness. Three miRNAs are siRNA-694, siRNA729 and siRNA-2259. We have rephrased this section as “The results showed that siRNA-694, siRNA729, and siRNA-2259 significantly reduced the relative expression level of NCAPG (P < 0.01) (Figure 3A), the interference efficiency excessed over than 70%, so siRNA-694, siRNA729 and siRNA-2259 can be used in subsequent experiments. Considering the interference efficiency of siRNA-694 is higher than other two siRNAs, siRNA-694 was used to the further experiment.” Please see line 341-344.
Q46: Line 338 - "siNRA-2259" → "siRNA-2259"
A46: Thank you very much for your careful work. We have revised “siNRA-2259” as “siRNA-2259”. Please see line 344.
Q47: Line 342 - "group was no difference (P > 0.05)" → "group was similar"
A47: Thank you very much for your valuable suggestion. We have revised "group was no difference (P > 0.05)" as "group was similar". Please see line 348.
Q48: Line 391 - " Pooled sanger sequencing was used to scan the potential SNPs in the NCAPG promoter region." → "Pooled Sanger sequencing was used to search for the potential SNPs in the NCAPG promoter region."
A48: Thank you very much for your valuable suggestion. We have revised "Pooled sanger sequencing was used to scan the potential SNPs in the NCAPG promoter region." As "Pooled Sanger sequencing was used to search for the potential SNPs in the NCAPG promoter region." Please see line 396.
Q49: Line 400 - "0.36, 0.36, 0.36, 0.33" - five values should be given (5 SNPs) or just range - 0.33-0.36
A49: Thank you very much for your careful work. We have rewrite this sentence as “The polymorphic information content of these five SNP was 0.36, 0.36, 0.36, 0.36 and 0.33 separately.” Please see line 406.
Q50: Section 3.6 - italicize genotypes e.g. CC and do not use phrase "biggest phenotype" just for example "individuals carried TT genotype were characterized by highest values of all/investigated/growth and developmental traits"
A50 Thank you very much for your valuable comments. We have rephrased this section as “In SNP1, individuals carried AA genotype were characterized by highest values of investigated traits. In SNP2, individuals carried TT genotype were characterized by highest values of investigated traits. In SNP3, individuals carried CC genotype were characterized by highest values of investigated traits. In SNP4, individuals carried TT genotype were characterized by highest values of investigated traits. In SNP5, individuals carried AA genotype were characterized by highest values of investigated traits.” Please see line 420-425.
Q51: Line 424 - "detected" → "analyzed"
A51: Thank you very much for your valuable suggestion. We have revised “detected” as “analyzed”. Please see line 434.
Q52: Line 445 - "increase" → " increased"
A52: Thank you very much for your valuable suggestion. We have revised “increase” as “increasee”. Please see line 454.
Q53: Line 447-450 - do not use capitals for cell cycle stages names
A53: Thank you very much for your valuable suggestion. We have used lowercase for cell cycle stages names. Please see line 454-459.
Q54: Line 452 - "Differentiation processes is different" - rewrite it, please
A54: Thank you very much for your suggestion. We have rewrite this sentence as “Differentiation is different from proliferation”. Please see line 461.
Q55: Line 454 - "An"→ "a"
A55: Thank you very much for your careful work. We have revised “An” as “A”. Please see line 463.
Q56: Line 456 – NCAPG
A56: Thank you very much for your careful work. We have revised gene name. Please see 465.
Q57: Line 462 - "Screen SNPs in" → "Screen for SNPs in"
A57: Thank you very much for your careful work. “for” has been added between “Screen” and “SNP” in text. Please see line 470.
Q58: Line 463 - " identify usefully genetic marker for breeding." → "identify useful genetic markers for animal breeding."
A58: Thank you very much for your valuable suggestion. “animal” has been added between “for” and “breeding”. Please see line 471.
Q59: Line 464 - "region of NCAPG associated" → "region of NCAPG were associated"
A59: Thank you very much for your valuable suggestion. “were” has been added between “NCAPG” and “associated”. Please see line 471
Q60: Line 471 - "A SNP' → "For example SNP"
A60: Thank you very much for your valuable suggestion. We have revised “A SNP” as “For example SNP”. please see line 479.
Q61: Line 475 - "to regulate" → "and influence"
A61: Thank you very much for your suggestion. We have revised “to regulated” as “and influence”. Please see line 483.